# Correction of Dynamical Properties of Data Acquisition Systems

**DOI:** 10.3390/s23031676

**Published:** 2023-02-03

**Authors:** Jarosław Figwer, Małgorzata I. Michalczyk

**Affiliations:** Department of Measurements and Control, Silesian University of Technology, Akademicka 16, 44-100 Gliwice, Poland

**Keywords:** measurement systems, data acquisition systems, smart sensors, dynamical correctors

## Abstract

Accurate and fast measurements are important in many areas of everyday engineering and research activities. This paper proposes a method that gives such measurements, additionally shortening the time in which they are obtained. To achieve this, a supplementary discrete-time filter, estimating values of delayed samples of the measured signal, is attached to the output of the data acquisition system. This filter is identified with the use of classical estimation methods, based on a dynamical model of the data acquisition system. The definition of the cost function minimised during filter identification depends on the nature of the environment in which measurements are acquired. The considerations presented in this paper are illustrated with four corresponding simulated case study examples. They show how, in a very simple way, dynamical properties of data acquisition systems may be corrected, and thus improved, using the corresponding supplementary discrete-time filters. The improvement, measured by the correction quality index introduced in the paper, was from a few times up to more than 100. The paper also raises the issue of obtaining models for tuning of the supplementary discrete-time filter. The considerations presented may be applied to formulate the artificial intelligence of data acquisition systems as well as sensors. Finally, the paper proposes to implement this intelligence as a knowledge base of the expert system.

## 1. Introduction

Data acquisition systems are an inherent part of modern digital signal processing systems met in everyday life. Their quality is determined by the ability to provide accurate results of measurements in a short period of time. It follows from the control theory that the length of time needed to obtain accurate measurements is bounded from below by dynamical properties of data acquisition systems. It was shown in [1] how dynamical properties of data acquisition systems may be on-line corrected to enhance the performance of active noise control systems. The idea was to attach to the output of the data acquisition system a supplementary discrete-time filter, that was used to estimate values of the delayed error signal samples at the input of the error microphone. Considerations presented below are a generalisation of this concept. Now, this idea is used to correct dynamical properties of general data acquisition systems for any measurements, acquired in both random and deterministic environments. This correction is obtained by attaching to the output of the data acquisition system a supplementary discrete-time filter estimating values of delayed samples of the measured signal. The supplementary discrete-time filter may be identified using classical estimation methods. In the choice of this filter structure, methods of model selection well known from identification theory, may help.

The proposed idea is new in the theory of measurements. The extensive literature search resulted in some publications correlated with the subject of this paper. The correction of dynamical properties of data acquisition systems and sensors is obtained there by inverse modelling [2,3,4,5,6,7,8], or inverse modelling aided by a feedback control system [9], or one-step forward specialised prediction [10,11,12], or joint input and state estimation based on Kalman filtering [13,14,15,16,17,18,19,20]. All these correction methods use linear discrete-time dynamic models of data acquisition systems. Because of the non-minimum phase property of data acquisition system models implied by their discretisation, it is difficult to determine the corresponding inverse models giving sufficiently accurate estimates of the measured signal. Additionally, single-step forward prediction is a less flexible tool than estimation of delayed measured signal values with the use of the supplementary discrete-time filter. While the forward prediction horizon is extended, the error of prediction increases. In contrast, extending of the corresponding delay in estimating measured signal values results in more accurate estimates, which is proven in this paper. Using the idea proposed in this paper, there is also no necessity to jointly estimate measured signal values and the state of the data acquisition system. It radically reduces numerical complexity of the obtained corrector (the supplementary discrete-time filter) and hence simplifies its technical implementation. The novelty of the proposed idea consists also in the use of continuous-time dynamic models of data acquisition systems in supplementary discrete-time filter design. Moreover, supplementary discrete-time filters identified in the design stage may be further changed during data acquisition system operation, according to time-varying properties of measured signals.

The paper consists of four parts. The first part describes in detail the proposed idea of correction of the dynamic properties of data acquisition systems. In the second part, four simulated case study examples are presented to show how the use of supplementary discrete-time filters may correct dynamical properties of data acquisition systems and sensors. Next, comments on obtaining models of data acquisition systems and sensors, illustrated by a practical example using data from real-world identification experiment, are included. Finally, the issues related to new ideas for defining knowledge bases of smart data acquisition systems as well as smart sensors are raised.

## 2. Basic Ideas

In this paper, a data acquisition system is a dynamical system allowing one to transform signals from the environment around us into numbers stored in a computer. The exemplary data acquisition system may consist of a sensor, amplifier, antialiasing filter and A/D converter containing a quantizer (Figure 1). It is assumed in the considerations presented below that the data acquisition system is a continuous-time dynamical system at the output of which an A/D converter is connected. The resulting system transforms measured signal y(t) from continuous-time domain *t* ($t\in R^{+}$) to the discrete-time domain u(iT), where *i* (i=0,1,…) denotes consecutive discrete-time instants and *T* is the sampling interval. The corresponding transformation of y(t) into u(iT) is a dynamic one—it cannot be expected that u(iT) will be equal to y(iT) for all discrete-time instants iT. This equality is possible only for y(t) being a constant value after all transients implied by dynamics of the data acquisition system have decayed. For y(t) varying with time values u(iT) are never equal to values of y(t) at the discrete time instants iT. To reduce discrepancy between these values, a supplementary discrete-time filter estimating values of delayed samples of the measured signal y(t) is attached to the output of the data acquisition system (Figure 1). Its output discrete-time signal is denoted by v(iT). The supplementary discrete-time filter can be identified during the design stage of the data acquisition system, using ideas of a routine for equalisation of communication transmission channels [21,22,23]. Its parameters—structure numbers and, for example, coefficients of the corresponding transfer function—can be obtained by minimisation of a cost function, which definition depends on the properties of the measured signal y(t). If the measured signal y(t) is a weak ergodic random process one of the following cost functions
(1)$$S_{1}\left(i,\Delta \right)=E\left\{{\left(y_{p}\left(iT-\Delta \right)-v_{p}\left(iT\right)\right)}^{2}\right\},$$
where $E\left\{\cdot \right\}$ is the expectation operator and i=0,1,…,N−1 or
(2)$$S_{2}\left(\Delta \right)=\sum_{i=0}^{N-1}{\left(y_{p}\left(iT-\Delta \right)-v_{p}\left(iT\right)\right)}^{2},$$
may be applied to obtain parameters of the supplementary discrete-time filter, that estimates (predicts) values of delayed samples of the measured signal. In the case, when the measured signal y(t) is a deterministic one, only the cost function $S_{2}\left(\Delta \right)$ can be used in calculation of the parameters of this filter. In the above definitions of cost functions, *N* is the number of processed signal samples in the design stage, Δ is a discrete-time delay, being a parameter of the supplementary discrete-time filter, $y_{p}\left(t\right)$ is a continuous-time signal simulating properties of the measured signal y(t) and $v_{p}\left(iT\right)$ is the response to the excitation $y_{p}\left(i\right)$ of the dynamic system, being a series connection of the dynamic continuous-time model of the data acquisition system and the supplementary discrete-time filter.

To perform minimisation of the above defined cost functions, the dynamic continuous-time model of the data acquisition system must be known. This model can be, for example, identified by the user or provided by the manufacturer of the data acquisition system. During minimisation of the above cost functions, the dynamical continuous-time model of the data acquisition system is excited by artificially generated weak ergodic or deterministic input signals $y_{p}\left(t\right)$. Properties of these signals depend upon the purpose for which the data acquisition system will be used. In the case of performing measurements of random phenomena, realisations of continuous-time multisine random signals with predefined spectral properties simulating properties of the acquired signals may be used as the signal $y_{p}\left(t\right)$. In the case of measurements in a deterministic environment, step signal, ramp function with saturation or triangular and square waves may be applied as the signal $y_{p}\left(t\right)$. It should be emphasised that properties of the signal $y_{p}\left(t\right)$ influence obtained parameters of the supplementary discrete-time filter—there is no one optimal supplementary discrete-time filter providing accurate measurements of all kinds of measured signals.

Obtaining of the supplementary discrete-time filter parameters (called tuning) may be realised via classical input–output plant identification methods using the simulated signal $u_{p}\left(iT\right)$, this being the output signal of the data acquisition system excited by $y_{p}\left(i\right)$ as the input and samples of the signal $y_{p}\left(iT-\Delta T\right)$ ($v_{p}\left(iT\right)=y_{p}\left(iT-\Delta T\right)$) as the output of the plant to be identified [24,25]—see Figure 2. It is especially simple when the supplementary discrete-time filter is a rational discrete-time filter with the transfer function $F_{est}\left(z^{-1}\right)$ ($z^{-1}$ is a one-step delay operator). For example, in the case of the supplementary discrete-time filter being a discrete-time FIR filter there are two design parameters: the discrete-time delay Δ and the degree dF of the discrete-time transfer function polynomial $F_{est}\left(z^{-1}\right)$. These design parameters define the structure of the discrete-time FIR filter, that being the supplementary discrete-time filter. In cases in which the supplementary discrete-time filter is a discrete-time IIR filter, three design parameters define its structure: the discrete-time delay Δ and the degrees dF and dA of the discrete-time transfer function $F_{est}\left(z^{-1}\right)$ numerator and denominator. When the linear supplementary discrete-time filter is not effective, more complicated filters being block-oriented nonlinear dynamic systems or neural networks may be applied.

## 3. Simulated Case Studies

The proposed idea of correction of dynamical properties of the data acquisition systems by attaching supplementary discrete-time filters is illustrated below with simulated case studies. In order to get insight into a problem, four case studies are designed, differing in the data acquisition system structure and also in its dynamical properties. First, a data acquisition system with neglected dynamical properties of the sensor is considered. Then, a sensor with embedded antialiasing filter is taken into account. The third case study concerns a data acquisition system without neglecting dynamical properties of a sensor and aliasing filter. These three exemplary systems are used to measure random phenomena assuming that the measured signals are weak ergodic random signals. In contrast, in the last simulated case study dynamical properties of a sensor used to measure a deterministic signal are corrected. Simulations are conducted with the following assumptions:Due to an easy graphical presentation of the results obtained in the design stage, FIR filters are applied as supplementary discrete-time filters. There are no restrictions with respect to using IIR filters as well.Special attention is paid to a supplementary filter design process, namely, concerning the choice of the input signals $y_{p}\left(t\right)$ used for estimation of $F_{est}\left(z^{-1}\right)$ coefficients. These signals should reflect dynamical properties of the acquired signals. In the case of the first three simulated case studies, they are chosen as one or many realisations of white (band limited to a determined frequency range) or coloured continuous-time multisine random (CTMR) signals of the period of $N_{p}T$ ($N_{p}=2N_{s}$), with $N_{s}$ having continuous-time harmonic sine components plus a constant component [26,27]. Their standard deviation is denoted by σ.The supplementary discrete-time filters are tuned—their coefficients are obtained by minimisation of the cost function $S_{1}\left(i,\Delta \right)$ or $S_{2}\left(\Delta \right)$—using ideas presented in [28,29]. In this operation, number of samples used *N* less than $N_{s}$ means that a part of the period of the CTMR signal is used; further, number *N* greater that $N_{s}$ means using circular extension of the CTMR signal.The following four operating modes of the A/D converter are considered (Figure 3):–Mode A/D NQ: perfect A/D signal conversion without the above-mentioned quantizer;–Mode A/D Q: A/D signal conversion with quantizer;–Mode A/D RQ: A/D signal conversion aided by randomised quantisation;–Mode A/D RF: A/D signal conversion aided by random two-bit fluctuations.Randomised quantisation involves adding to processed signal values, prior to quantisation, independent realisations of a random variable, uniformly distributed in the range covering the data acquisition system quant [30]. Random two-bit fluctuations are realised similarly by adding, prior to quantisation, realisations of a random variable uniformly distributed in a range covering two quants of the quantizer used.

In the paper, the quality of the data acquisition system with the attached supplementary filter $F_{est}\left(z^{-1}\right)$, obtained as a result of minimisation of the cost function $S_{1}\left(i,\Delta \right)$ or $S_{2}\left(\Delta \right)$, is assessed with the use of the following correction quality index
(3)$$C\left(\Delta ,dF\right)=\frac{\frac{1}{N}\sum_{i=0}^{N-1}{\left({\bar{v}}_{p}\left(iT\right)-y_{p}\left(iT-\Delta T\right)\right)}^{2}}{\frac{1}{N}\sum_{i=0}^{N-1}{\left(u_{p}\left(iT\right)-y_{p}\left(iT\right)\right)}^{2}}100\%,$$
where ${\bar{v}}_{p}\left(iT\right)$(i=0,1,…,N−1) are values on the output of the supplementary discrete-time filter $F_{est}\left(z^{-1}\right)$, for $F_{est}\left(z^{-1}\right)$ coefficients calculated in such a way as to provide a minimum of the cost function for the given signal $y_{p}\left(t\right)$. Values of this index below 100% show an increase in accuracy of the measurements obtained by the data acquisition system after attaching the supplementary discrete-time filter. The lower the correction quality index value, the better the correction of the dynamical properties of the data acquisition system.

In the simulated case studies presented below, there is no systematic comparison of the proposed idea with literature methods of correcting dynamical properties of data acquisition systems.

### 3.1. Data Acquisition System with Neglected Dynamics of a Sensor

In the first simulated case study example, a supplementary discrete-time filter was designed for use with the data acquisition system containing only one dynamic component, which was an antialiasing filter realised as the analogue Butterworth filter of order 4 with cutoff frequency equal to 140.7600 Hz, as shown in block diagrams in Figure 4. In Figure 5, the magnitude of its frequency response is shown. It was assumed that the transfer function of the sensor attached to the input of this data acquisition system was equal to 1.0000. The data acquisition system was working with the sampling interval *T* equal to 0.0020 s.

Coefficients of considered supplementary discrete-time filters $F_{est}\left(z^{-1}\right)$ of different structures (different values of parameters Δ and dF, where Δ≤dF) were obtained by minimisation of the cost function $S_{1}\left(i,\Delta \right)$, for each time instant *i*. For each structure of the supplementary discrete-time filter, the minimisation was repeated for 100 pattern realisations of the signal $y_{p}\left(t\right)$, with the band limited to the range [0,250] Hz white CTMR signals with standard deviation σ=1.0000 and $N_{s}=$ 1,000,200 [26,27] (Figure 4a). In Figure 6, the corresponding periodogram of each simulated realisation of the length $N_{p}=$ 2,000,400 samples is shown. The same signal realisations were used to calculate values of the correction quality index C(Δ,dF) for the chosen values of Δ and dF, assuming N= 1,000,200 (a half of the period of the signal $y_{p}\left(t\right)$) (according to block diagram Figure 4c). It follows from the obtained mean correction quality index values C(Δ,dF), presented in Figure 7, that attaching to the output of the data acquisition system the supplementary discrete-time filter of any considered structure increases accuracy of measurements. Moreover, for dF greater than 6 and Δ in the range (2,dF−1), all calculated mean values of the correction quality index C(Δ,dF) are less than 1.0000%.

The above-described simulation experiment was repeated for a single realisation of a coloured CTMR signal with σ=1.0000, $N_{s}=$ 500,000 and $N_{p}=$ 1,000,000 (Figure 4b). In Figure 8, a periodogram of this realisation is shown. It was assumed that *N* is equal to 1,000,000 and the sampling interval T=0.0020 s. In Figure 9, the corresponding results of the correction quality index C(Δ,dF) calculated as in Figure 4d are presented. Again, attaching the supplementary discrete-time filter of any considered structure to the output of the data acquisition system increases accuracy of measurements. Further, for dF greater than 11 and Δ in the range (4,dF−1), all calculated mean values of the correction quality index C(Δ,dF) are less than 1.0000%.

Thus, it turns out that both for white and coloured CTMR signals measured via data acquisition systems a relatively low degree dF of the supplementary discrete-time filter and low Δ assure large improvement in accuracy of measurements. It is worth emphasising that while using the presented idea in measurements, in which there is no need to limit time for decision-making, large Δ and dF can be used. In contrast, while choosing parameters (Δ and dF) of filter $F_{est}\left(z^{-1}\right)$ for application of this idea in control, there is a trade-off to be reached between the increase in the accuracy of the measurements and the required speed of the control system response. Although the increase in Δ and dF introduces additional delays in the estimation of measured signal values, the improvement in accuracy of measurements results in faster transient response of the control system [1].

In the last step of this simulated case study example (Figure 4e,f) coefficients of the supplementary discrete-time filter with dF=14 and Δ=7 were calculated for single realisations of the above-defined band limited to the range [0,250] Hz white and coloured CTMR signals ($N_{s}=$ 500,000 and N= 1,000,000). The corresponding calculated values of the correction quality index C(Δ,dF) were less than 1.0000%. Next, the supplementary discrete-time filter tuned using the band limited to the range [0,250] Hz white CTMR signal was attached to the data acquisition system, which was excited by 100 realisations of the coloured CTMR signal (Figure 4e). Subsequently, the calculated mean value of the correction quality index C(Δ,dF) was equal to 16.8900%. The inverse calculations (Figure 4f) resulted in the mean value of C(Δ,dF) equal to 62.4300%. It follows from these calculations that:Both supplementary discrete-time filters increase the accuracy of the considered data acquisition system, regardless of the properties of the measured signal;The supplementary discrete-time filter tuned for the band limited to the range [0,250] Hz white CTMR signal gave more accurate results of measurements of coloured CTMR signal than the corresponding supplementary discrete-time filter tuned for coloured CTMR signal and then applied to measure band limited to the range [0,250] Hz white CTMR signal;The greatest possible increase in the data acquisition system accuracy, in the sense of the correction quality index values, can be obtained for the corresponding supplementary discrete-time filters tuned using signals with spectral properties similar to those, which are exhibited by measured signals.

### 3.2. Sensor with Embedded Antialiasing Filter

The second simulated case study example concerns the design of a corrector for the data acquisition system containing only a sensor with dynamical properties defined by the following transfer function:(4)$$K\left(s\right)=\frac{6.0000}{0.8000s^{2}+3.4000s+6.000},$$
where s=jω, $j^{2}=-1$ and ω is the angular frequency. In Figure 10, the magnitude of its frequency response is presented. There is no additional antialiasing filter attached to the sensor in this data acquisition system—the antialiasing filter is embedded into the sensor. Only 16-bit A/D converter with saturation at levels −5.0000 and 5.0000 V is connected at the output of the sensor. It works with the sampling interval *T* equal to 0.1000 s.

In Figure 11, values of the correction quality index C(Δ,dF) calculated for supplementary discrete-time filters used to correct dynamical properties of the sensor with embedded antialiasing filter are presented. The structure of supplementary filters was arbitrarily selected as Δ=8 and dF=18. The corresponding filters’ coefficients were obtained by minimisation of the cost function $S_{1}\left(\Delta ,dF\right)$ for $y_{p}\left(t\right)$ being a single realisation of a band limited to the range [0,5] Hz white CTMR signal with standard deviation equal to σ, $N_{s}=$ 250,000 and N= 1,000,000, according to the tuning setup in Figure 12a. Calculations of the correction quality index C(Δ,dF) (experiment setup in Figure 12c) were repeated for 367 values of standard deviation σ nonuniformly distributed in the range $[10^{-5},$20,000] considering A/D converter operating modes: A/D NQ (green line in Figure 11), A/D Q (red line in Figure 11), A/D RQ (blue line in Figure 11) and A/D RF (magenta line in Figure 11). It follows from the presented results that calculated values of the correction quality index are:Less than 100% for all standard deviations σ considered;Less than 1% (solid black line in Figure 11) for standard deviation σ in the approximate range $\left[2\cdot 10^{-2},4\right]$;Similar for all operating modes of the A/D converter for standard deviation σ bigger than $2\cdot 10^{-2}$;The smallest for A/D Q mode if signal standard deviation σ is bigger than $2\cdot 10^{-4}$;The smallest for A/D RF mode considering standard deviations σ less than $2\cdot 10^{-4}$.

In the next step, coefficients of four supplementary discrete-time filters were calculated using the band limited to the range [0,5] Hz white CTMR signal with the standard deviation σ=1.0000 (tuning setup in Figure 12b). They were obtained for all the above-mentioned operating modes of the A/D converter by minimising the cost function $S_{1}\left(i,\Delta \right)$. Each supplementary discrete-time filter considered was then attached to the output of the data acquisition system with A/D converter working in four considered modes. For 367 values of standard deviation σ nonuniformly distributed in the range $[10^{-5},$20,000] and each supplementary discrete-time filter with the coefficients calculated for the standard deviation σ=1.0000, the corresponding values of correction quality index were calculated (experiment setup in Figure 12d). They are represented in Figure 13 (linear scale of correction quality index values) and in Figure 14 (logarithmic scale of correction quality index values) by dashed lines, while solid lines represent results obtained for supplementary discrete-time filters tuned for all values of $\sigma \in [10^{-5},$20,000], respectively. It follows from the results presented in these figures that for the measured signals with standard deviation σ in the approximative range $\left[10^{-2},10\right]$ one supplementary discrete-time filter tuned for σ=1.0000 behaves similarly to supplementary discrete-time filters tuned, respectively, for all consecutive values of σ.

In the last experiment of this simulated case study example (Figure 12e), a supplementary discrete-time filter, with coefficients calculated in the manner described above for the A/D NQ operating mode (Figure 12b), was attached to the data acquisition system with A/D converter exhibiting random *Q*-bit fluctuations. Random *Q*-bit fluctuations were modelled by adding to the signal processed, prior to quantisation, realisations of a random variable uniformly distributed in a range covering *Q* quants of the quantizer used. As in the previous simulation experiments for 367 values of standard deviations σ nonuniformly distributed in the range [10${}^{-5}$, 20,000] and Q=0,
1,
2,
5,
10,
20,
50,
100,
200,
500,
1000,
2000,
5000, 10,000, and 20,000, the corresponding values of the correction quality index were calculated. In Figure 15, obtained results are summarised. They prove great robustness of the proposed correction method of dynamical properties of data acquisition systems to random bit fluctuations. It is obvious that more precise measurements may be obtained by tuning the supplementary discrete-time filter with incorporation, in this tuning, of a knowledge about random bit fluctuations.

### 3.3. Case of Correction without Neglected Dynamical Properties of a Sensor and an Antialiasing Filter

In the third simulated case study example, a sensor with the following transfer function
(5)$$K\left(s\right)=\frac{1.0000}{sL+1.0000}$$
was attached to the input of an antialiasing filter that was the analogue Butterworth filter of order 4 with cutoff frequency 1.0000 Hz. The output of this filter was next processed via a 16-bit A/D converter working in the mode A/D Q with saturation at levels −5.0000 and 5.0000 V, with the sampling interval *T* equal to 0.1000 s. In Figure 16, the corresponding magnitude frequency responses of the sensor and antialiasing filter are presented considering different dynamical properties of the sensor, i.e., *L* equal to 0.0100, 0.0500, 0.1000 and 0.5000. It is worth noting that the passband of the sensor is wider than the passband of the antialiasing filter for *L* equal to 0.0100, 0.0500 and 0.1000. For L=0.5000, an opposite situation arises—the passband of the sensor is much narrower than the passband of the antialiasing filter.

A single realisation of a band limited to the range [0,5] Hz white CTMR signal with the standard deviation σ=1.0000 and $N_{s}=1024$ was generated as a signal $y_{p}\left(t\right)$. On this basis, coefficients of the following three supplementary discrete-time filters with arbitrarily selected structures were calculated via minimisation of the cost function $S_{2}\left(\Delta \right)$ (Figure 17):The supplementary discrete-time filter no. 1 (attached to the output of the sensor) with:
–The structure Δ=4, dF=25 for L=0.0100;–The structure Δ=1, dF=7 for L=0.0500;–The structure Δ=2, dF=15 for L=0.1000;–The structure Δ=8, dF=50 for L=0.5000;The supplementary discrete-time filter no. 2 (attached to the output of the antialiasing filter equipped with the 16-bit A/D converter) of the structure Δ=7, dF=140 for all considered values of *L*;The supplementary discrete-time filter no. 3 (attached to the output of the data acquisition system consisting of the sensor and antialiasing filter equipped with the 16-bit A/D converter) with:–The structure Δ=11, dF=165 for L=0.0100;–The structure Δ=8, dF=147 for L=0.0500;–The structure Δ=9, dF=155 for L=0.1000;–The structure Δ=15, dF=190 for L=0.5000.

Next, the following five dynamical systems were considered (Figure 18):System no. 1—the sensor with the attached supplementary discrete-time filter no. 1;System no. 2—the antialiasing filter with the attached supplementary discrete-time filter no. 2;System no. 3—the data acquisition system consisting of the serially connected sensor and antialiasing filter with the attached serially connected two supplementary discrete-time filters no. 2 and 1, respectively;System no. 4—the data acquisition system consisting of the sensor and antialiasing filter with the attached supplementary discrete-time filter no. 3;System no. 5—the data acquisition system consisting of the sensor and antialiasing filter with the attached supplementary discrete-time filter no. 2.

These systems were excited by 100 realisations of a band limited to the range [0,5] Hz white CTMR signal with the standard deviation σ=1.0000, $N_{s}=$10,000,000. In Table 1, mean values of the correction quality index calculated for these experiments and $N=2N_{s}$ are presented. It follows from the results presented in this table that:The supplementary discrete-time filter used like in the system no. 5 increases accuracy of the data acquisition systems considered for all values of *L*;The supplementary discrete-time filters used as in system no. 3 increase the accuracy of the data acquisition systems considered, except the value of *L* equal to 0.5000;The supplementary discrete-time filter used as in system no. 4 increase accuracy of data acquisition systems considered for all values of *L*, resulting in the smallest values of the correction quality index;For data acquisition systems containing sensors with the passband wider than the passband of antialiasing filter (L=0.01000 and L=0.0500), values of the correction quality index calculated for systems no. 3 and 4 are comparable;For the data acquisition system containing sensors with a passband narrower than the passband of the antialiasing filter, the value of the correction quality index calculated for system no. 4 is much smaller than the corresponding value calculated for system no. 5.

### 3.4. Measurements of Deterministic Signals

A data acquisition system, considered in the fourth simulated case study example, contained a sensor and an A/D converter without quantizer, working with the sampling interval *T* equal to 0.1000 s. The dynamical properties of the sensor were defined by the following transfer function
(6)$$K_{s}\left(s\right)=\frac{1.0000}{0.1000s^{3}+1.0000s^{2}+1.5000s+1.0000}.$$

The system was used to measure a signal changing like a step function. For the signal $y_{p}\left(t\right)$ as the unit step that appeared at the time instant 40.0000 s and lasted up to the time instant 409.6000 s (N=4096), the cost function $S_{2}\left(\Delta \right)$ was minimised to obtain coefficients of the supplementary discrete-time filter with dF equal to 39 (tuning setup in Figure 19a). In Table 2, the minimum values of the cost function $S_{2}\left(\Delta \right)$ obtained for Δ=1,2,…,10 are reported. Next, dynamical properties of the sensor were corrected by the supplementary discrete-time filter tuned this way (according to the experiment setup in Figure 19c). In Figure 20 and Figure 21, the response of the sensor to the unit step is compared with the corresponding step responses of the sensor with the attached supplementary discrete-time filter. For values of the delay Δ greater than 4 the supplementary discrete-time filter attached to the sensor reproduces accurately the delayed unit step. It should be emphasised that using the idea proposed in this paper dynamical properties of data acquisition systems may be tuned to any dynamical pattern defined by the signal $y_{p}\left(t\right)$. This allows for shaping dynamical properties of data acquisition systems.

In the next simulation experiment (Figure 19), values of the correction quality index C(Δ,dF) were calculated for two cases A and B (Table 2). In case A, the supplementary discrete-time filter was tuned for $y_{p}\left(t\right)$ being the unit step (Figure 19a). Then, this filter was attached to the sensor and the so obtained data acquisition system was excited by a realisation of the band limited to the range [0,5] Hz white CTMR signal with $N_{s}=2048$ and standard deviation σ=1.0000, and the correction quality index C(Δ,dF) was calculated for N=4096 (Figure 19d). In case B, coefficients of the supplementary discrete-time filter were obtained using a single realisation of this band limited to the range [0,5] Hz white CTMR signal as the signal $y_{p}\left(t\right)$ (Figure 19b). Next, the 18-bit A/D converter, processing data from the range [−5.0000,5.0000] V, was attached to the output of the sensor. For the so obtained data acquisition system (Figure 19e), the correction quality index C(Δ,dF) was calculated for the measured signal y(t) being a realisation of the same random signal assuming N=4096. It follows from the results presented in Table 2 that properties of the measured phenomena influence the accuracy of the data acquisition system with the attached supplementary discrete-time filter. The supplementary discrete-time filter that measures random signals and is tuned for the unit step change of a measured signal gives worse results than the supplementary discrete-time filter tuned for a random signal.

In the above-described simulated case study examples, supplementary discrete-time filters were discrete-time FIR filters. Similar results can be obtained for the supplementary discrete-time filters being discrete-time IIR filters. For example, repetition of calculations presented in this simulation example for the supplementary discrete-time IIR filter with the structure dF=7 and dA=2 and the unit step of $y_{p}\left(t\right)$ resulted, for Δ=5, in the minimum value of the cost function $S_{2}\left(\Delta \right)$ equal to $0.9\cdot 10^{-2}$, close to the corresponding value presented in Table 2.

## 4. Obtaining Models of Data Acquisition Systems

In order to apply the idea proposed in this paper and correct the dynamical properties of data acquisition systems, it is necessary to know the dynamical models of these systems. They can be provided by their manufacturers or identified by their users. In order to identify models, users of data acquisition systems need to choose an excitation signal that is preferably easy to implement. These can be deterministic signals, such as a unit step, a similar step or ramp function with saturation, as well as triangular and square waves. Next, it is worth using identification methods dedicated to these excitations [29,31].

To illustrate the process of model identification of such a system, a dynamical model of the data acquisition system containing Analog Devices low voltage temperature sensor TMP36gz with attached Texas Instruments 16-bit A/D converter ADS1115 was identified. The system worked with the sampling frequency 800 Hz. During the identification experiment, the sensor was very quickly moved from water at 0.0000
${}^{{}^{\circ}}$C to boiling water (100.0000 ${}^{{}^{\circ}}$C). It was assumed that this step excitation was introduced at the time instant 43.7500 s after the start of the identification experiment. The identification experiment lasted 241.6000 s. The samples obtained at the output of the data acquisition system were used to identify a continuous-time linear dynamic model of this system. This model was a nonrational transfer function, which was a serial connection of a part modelling a pure time delay with a rational part with the degrees of numerator and denominator polynomials equal to 1 and 2, respectively. The rational part of the identified data acquisition system model is described by the transfer function:(7)$$K\left(s\right)=\frac{1.9118\;\cdot \;10^{3}s+1.6035\;\cdot \;10^{2}}{2.3609\;\cdot \;10^{2}s^{2}+41.6143s+1.6720}.$$
Estimation of the time delay resulting in the value 1.2767 s. In Figure 22, the step excitation used for model identification, the output of the identified model and the measured step response of the data acquisition system are presented. It is worth emphasising that the sensor (or data acquisition system) calibration may be a part of the model identification procedure, but in the presented example it was not performed.

## 5. Intelligence of Data Acquisition Systems and Sensors

The above-presented simulated case study examples show that the choice of supplementary discrete-time filters used to correct dynamical properties of data acquisition systems depend on measured phenomena. To choose an adequate supplementary discrete-time filter structure and coefficients, the information about the properties of the measured signal, e.g., standard deviation or its power spectral density, is to be taken into account. This knowledge can be expressed as an additional rule-based knowledge base [32], which could be embedded into the data acquisition system to aid the choice.

Rules of the knowledge base may be initially used in tuning supplementary discrete-time filters based on an approximate definition of the properties (standard deviation, power spectral density) of the measured phenomena. For example, user of the data acquisition system may sketch a power spectral density of the signal measured. Next, using this information, the structure and coefficients of the corresponding supplementary discrete-time filter may be taken from the corresponding database, or they can be calculated based on the known or initially measured properties of the measured signal y(t). This calculation may be aided by a method of synthesis and simulation of random time-series with predefined spectral properties [26,33] and methods automating model structure selection [34]. This procedure, utilising the knowledge base, may also be applied to on-line calculation and update of supplementary discrete-time filters for the case of signals with time-varying properties.

## 6. Summary

In the paper, a new method to correct dynamical properties of a data acquisition system based on an estimation of delayed signal values at its input by using a specially designed supplementary discrete-time filter was proposed. This supplementary discrete-time filter, identified in the design stage, may be further changed during data acquisition system operation, in accordance with properties of a measured signal changing in time. The proposed method allows for building a new generation of data acquisition systems and sensors, that provide accurate measurements faster. Furthermore, a new look at defining knowledge bases for smart data acquisition systems as well as smart sensors was introduced. The presented considerations were illustrated with simulated case study examples that showed great effectiveness of the proposed method. The improvement of the data acquisition system accuracy obtained, measured via the correction quality index introduced in the paper, was from a few times up to more than 100. Finally, it is worth emphasising that although the discussion presented in the paper concerns linear data acquisition systems and sensors, the problem of correcting of dynamical properties of nonlinear data acquisition systems and sensors may be solved in the same way. In that case, supplementary discrete-time digital filters are nonlinear dynamical systems. Their tuning is an optimisation problem in which the cost functions introduced in the paper are minimised. The ideas presented may also be availed to shape dynamical properties of data acquisition systems.

## Figures and Tables

**Figure 1 sensors-23-01676-f001:**
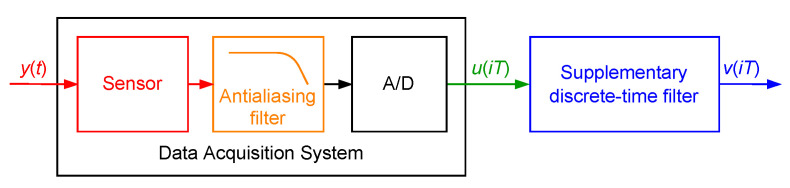
Data acquisition system with the attached supplementary discrete-time filter.

**Figure 2 sensors-23-01676-f002:**
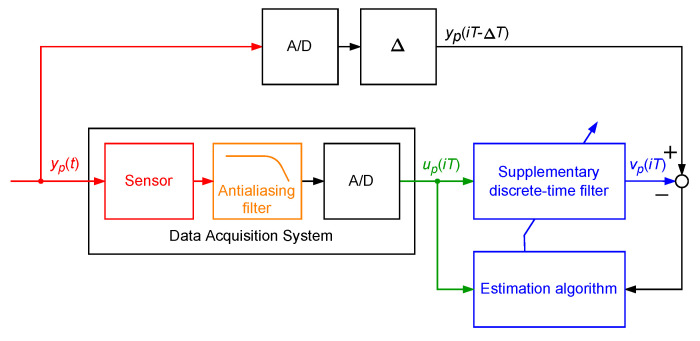
Tuning of the attached supplementary discrete-time filter.

**Figure 3 sensors-23-01676-f003:**
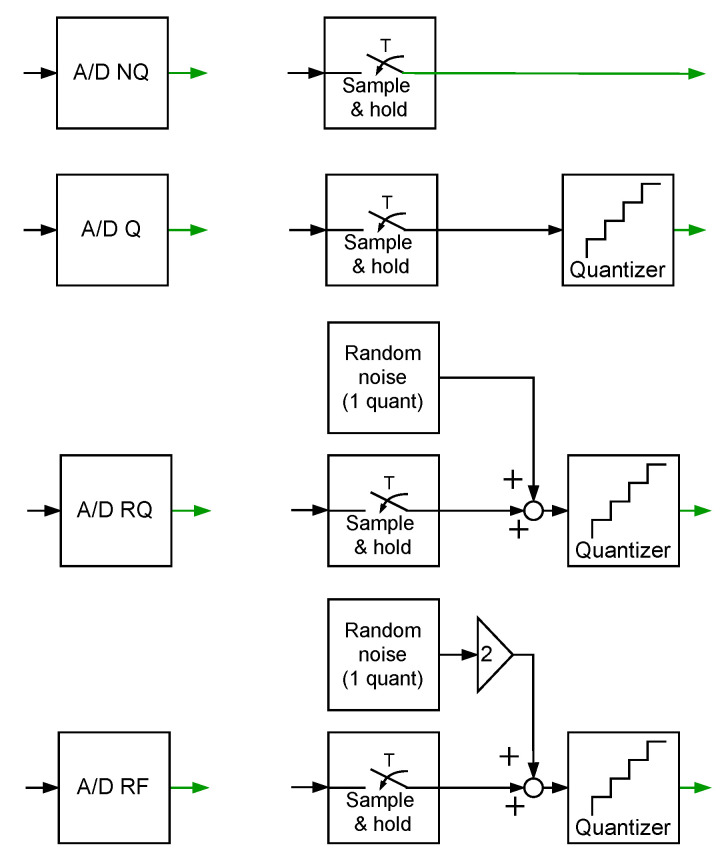
Operating modes of A/D converter in simulation experiments.

**Figure 4 sensors-23-01676-f004:**
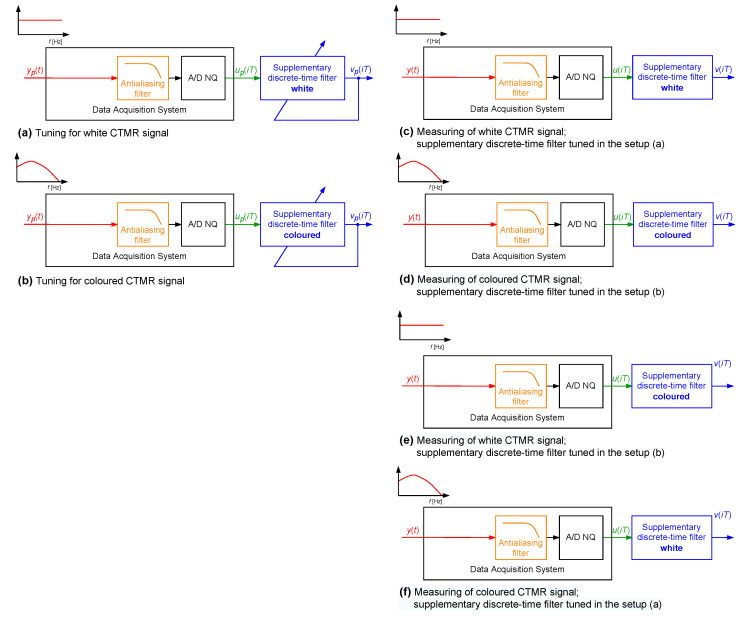
Setup of a simulation experiment—case of a data acquisition system with neglected dynamics of a sensor.

**Figure 5 sensors-23-01676-f005:**
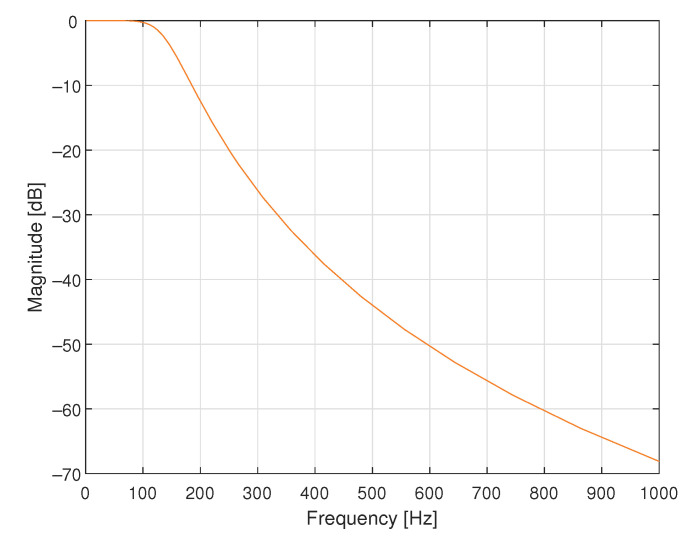
Magnitude of the frequency response of the analogue Butterworth filter of order 4 and cutoff frequency equal to 140.7600 Hz.

**Figure 6 sensors-23-01676-f006:**
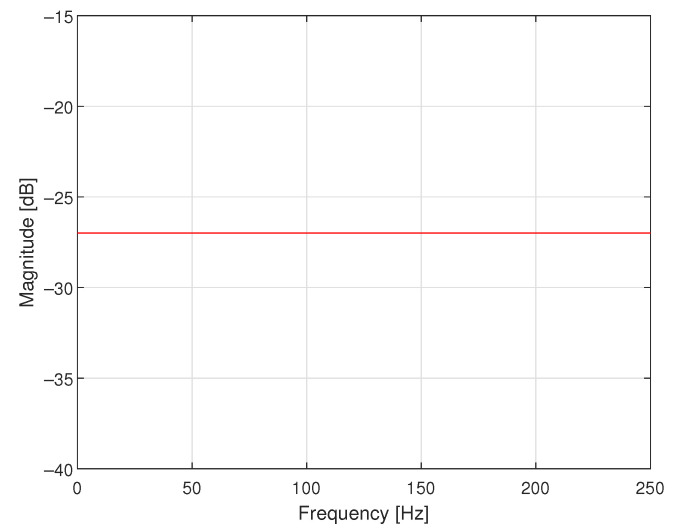
Periodogram of each realisation of a band limited to the range [0,250] Hz white CTMR signal with σ=1.0000, $N_{s}=$ 1,000,200, $N_{p}=$ 2,000,400 and T=0.0020 s.

**Figure 7 sensors-23-01676-f007:**
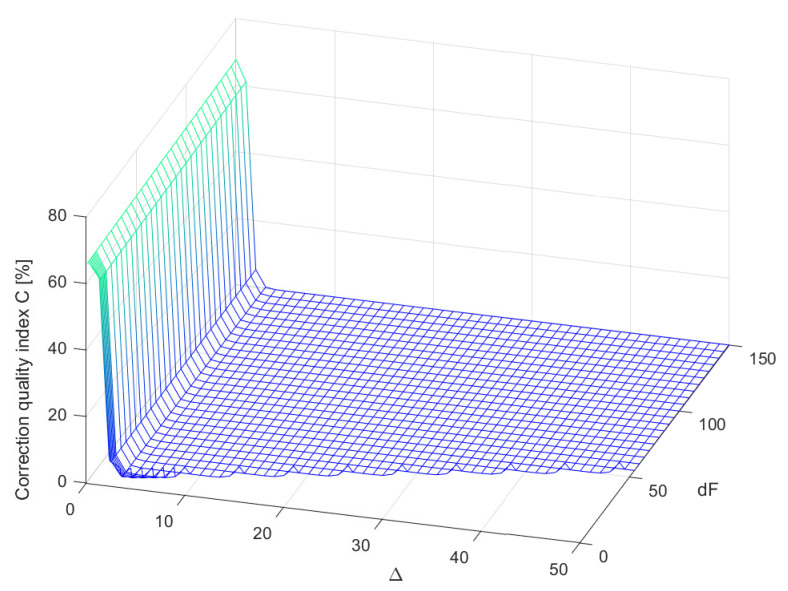
Mean values of the correction quality index C(Δ,dF) calculated for different structures of supplementary discrete-time filter, for 100 realisations of a band limited to the range [0,250] Hz white CTMR signal with σ=1.0000, $N_{s}=$ 1,000,200 and N= 1,000,200.

**Figure 8 sensors-23-01676-f008:**
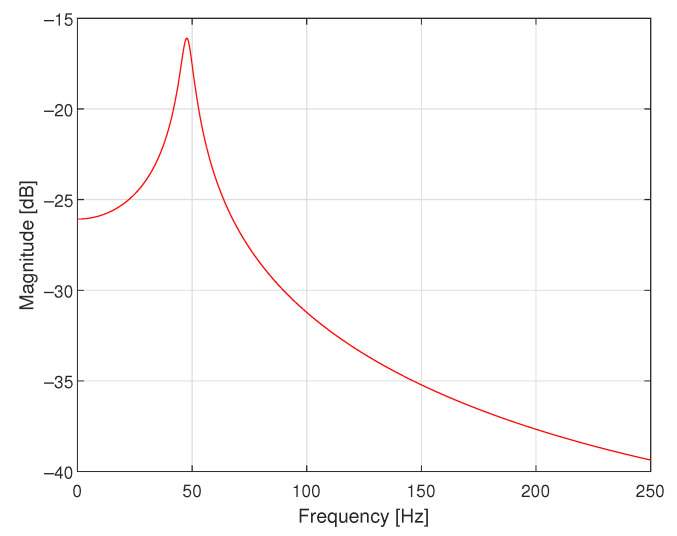
Periodogram of a single realisation of the coloured CTMR signal with σ=1.0000, $N_{s}=$ 500,000, $N_{p}=$ 1,000,000 and T=0.0020 s.

**Figure 9 sensors-23-01676-f009:**
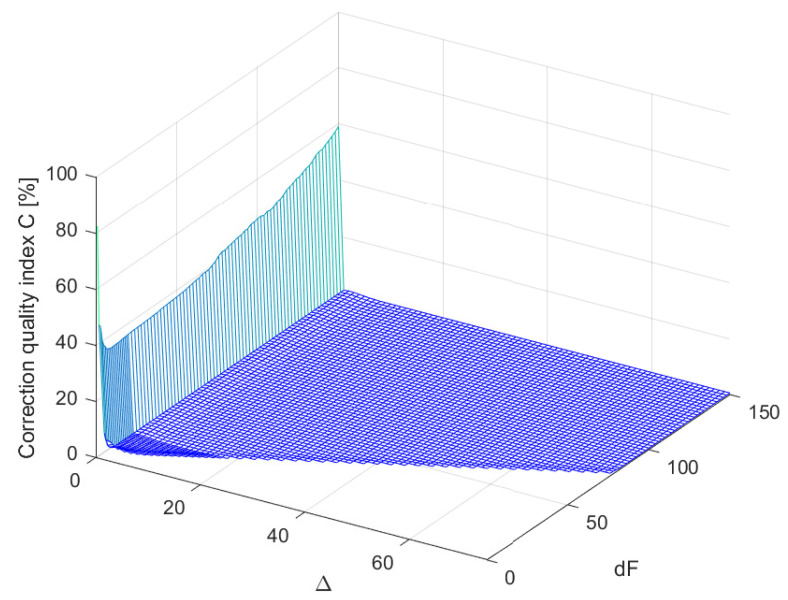
Mean values of the correction quality index C(Δ,dF) calculated for different structures of the supplementary discrete-time filters for a single realisation of a coloured CTMR signal with σ=1.0000, $N_{s}=$ 500,000 and N= 1,000,000.

**Figure 10 sensors-23-01676-f010:**
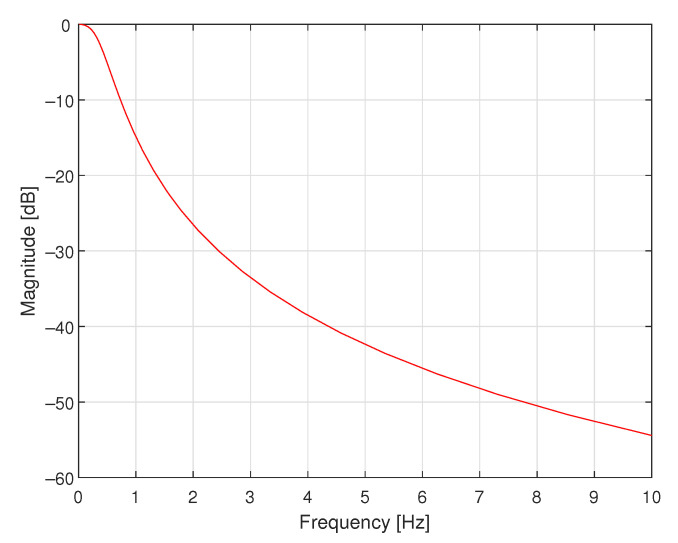
Magnitude of the frequency response of the sensor with embedded antialiasing filter.

**Figure 11 sensors-23-01676-f011:**
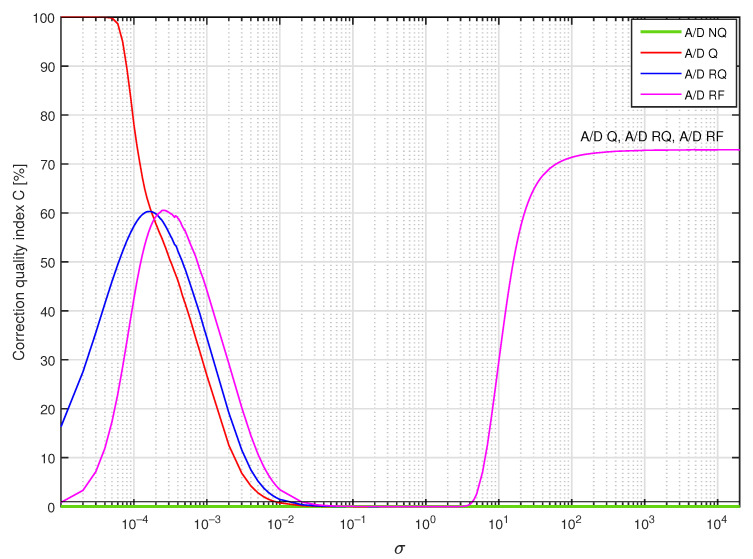
Values of the correction quality index C(8,18) calculated for supplementary discrete-time filters correcting dynamical properties of the sensor for a single realisation of a band limited to the range [0,5] Hz white CTMR signal with standard deviation σ, $N_{s}=$ 250,000 and N= 1,000,000.

**Figure 12 sensors-23-01676-f012:**
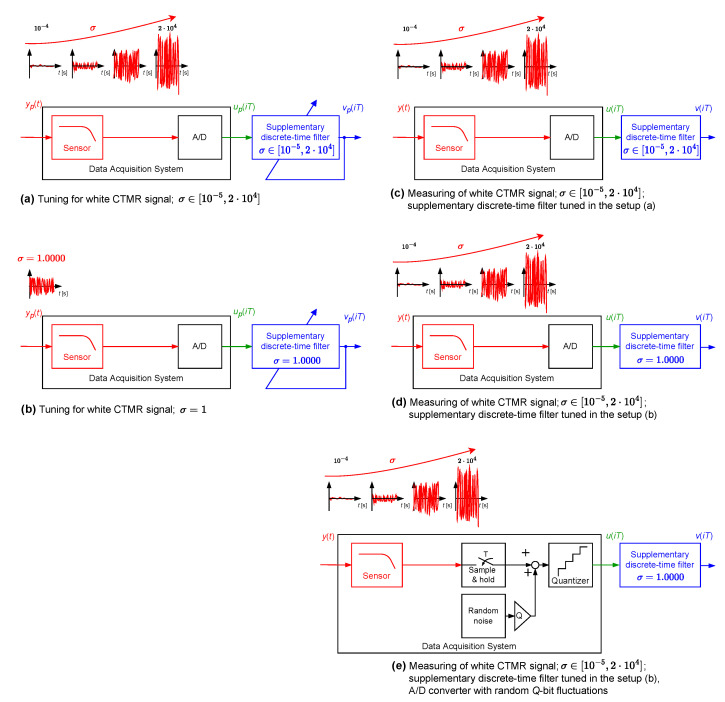
Setup of a simulation experiment—case of the sensor with the embedded antialiasing filter.

**Figure 13 sensors-23-01676-f013:**
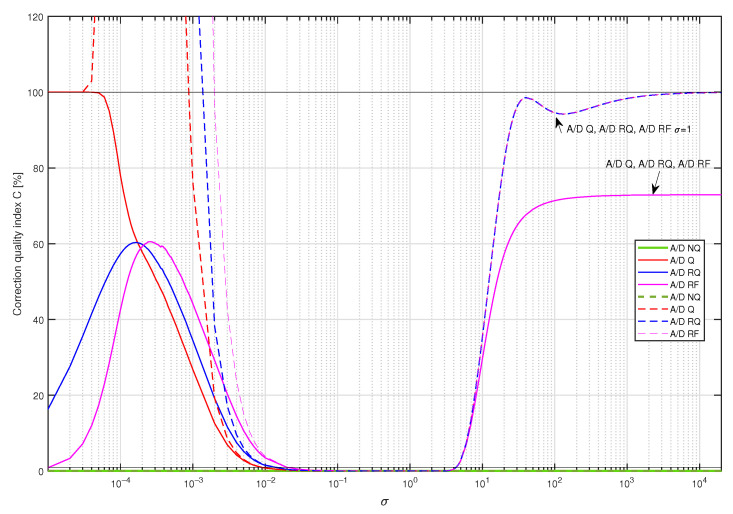
Values of the correction quality index C(8,18) (linear scale) calculated for supplementary discrete-time filters correcting properties of the sensor for a single realisation of a band limited to the range [0,5] white CTMR signal with standard deviation σ, $N_{s}=$250,000 and N=1,000,000. Solid lines present results for supplementary discrete-time filters tuned for all values of σ, while dashed lines present results for the corresponding filters tuned only for σ=1.0000.

**Figure 14 sensors-23-01676-f014:**
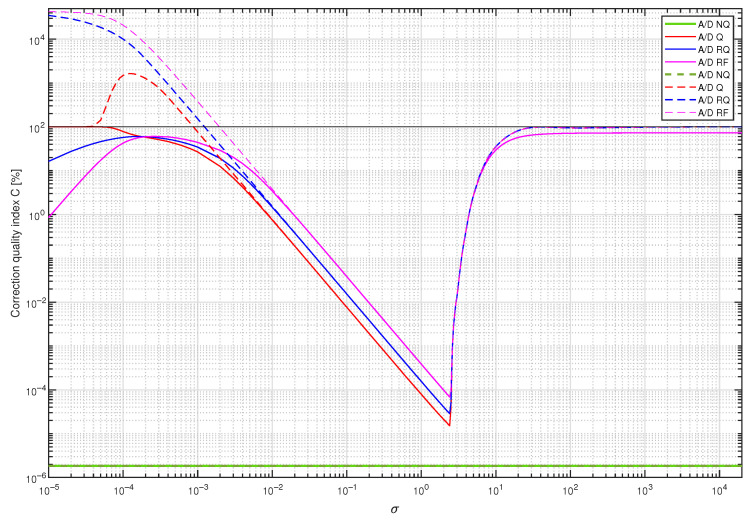
Values of the correction quality index C(8,18) (logarithmic scale) calculated for supplementary discrete-time filters correcting properties of the sensor for a single realisation of a band limited to the range [0,5] white CTMR signal with standard deviation σ, $N_{s}=$ 250,000 and N= 1,000,000. Solid lines present results for supplementary discrete-time filters tuned for all values of σ, while dashed lines present results for the corresponding filters tuned only for σ=1.0000.

**Figure 15 sensors-23-01676-f015:**
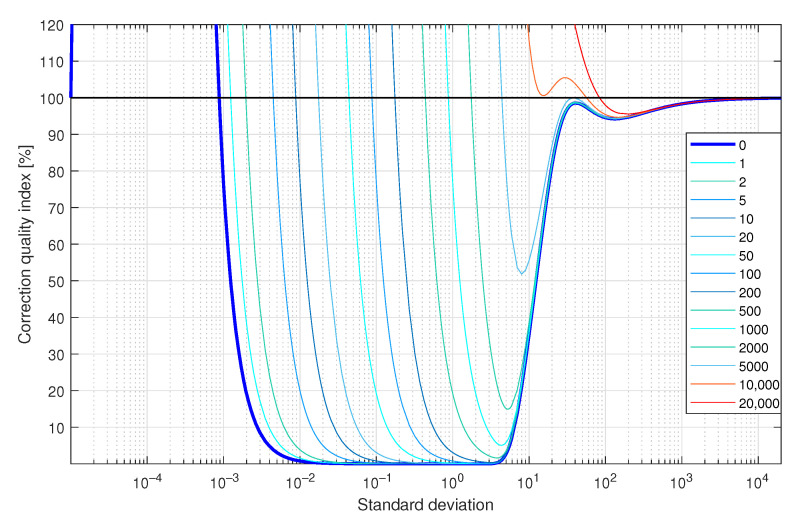
Values of the correction quality index C(8,18) (linear scale) calculated for the supplementary discrete-time filter correcting properties of the sensor for a single realisation of a band limited to the range [0,5] white CTMR signal with values of standard deviation σ, $N_{s}=$ 250,000, N=1,000,000 and random bit fluctuations covering number of quants listed in the legend—results obtained for the supplementary discrete-time filter tuned for σ=1.0000.

**Figure 16 sensors-23-01676-f016:**
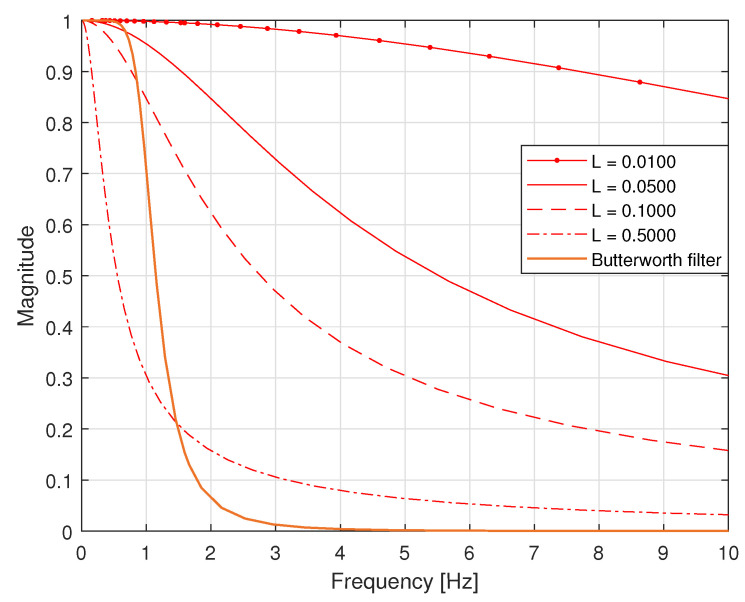
Magnitude of the frequency responses of sensors and the Butterworth filter.

**Figure 17 sensors-23-01676-f017:**
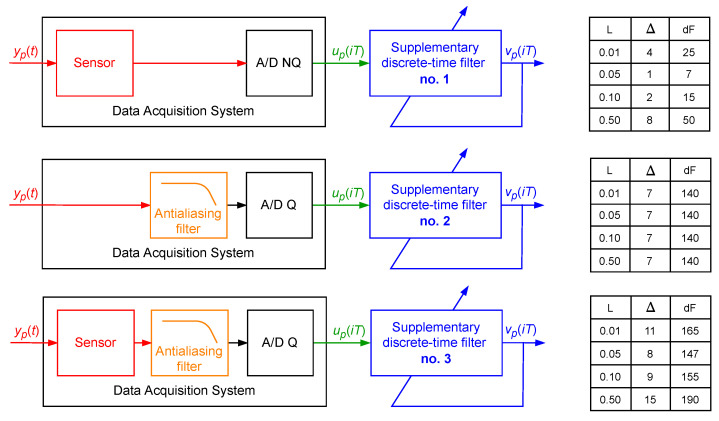
Setups for tuning of the supplementary discrete-time filters no. 1, 2, 3.

**Figure 18 sensors-23-01676-f018:**
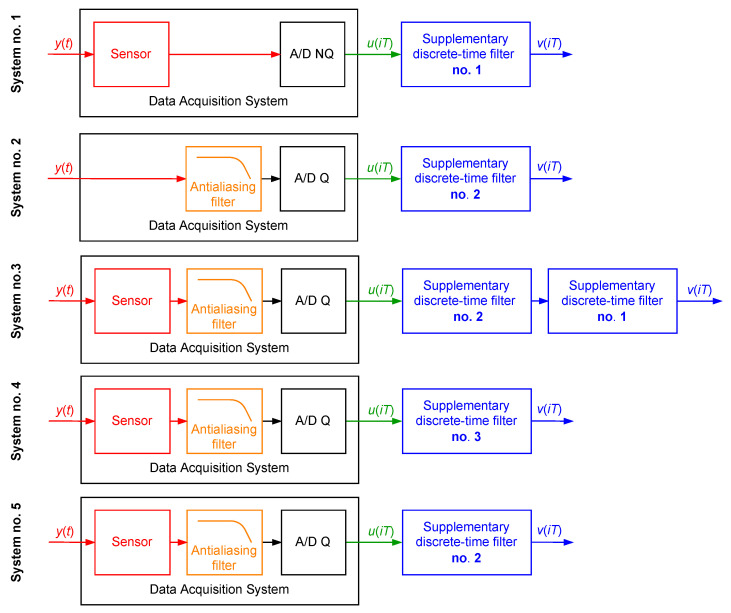
Setups of the supplementary discrete-time filter application in the simulation case study 3.3.

**Figure 19 sensors-23-01676-f019:**
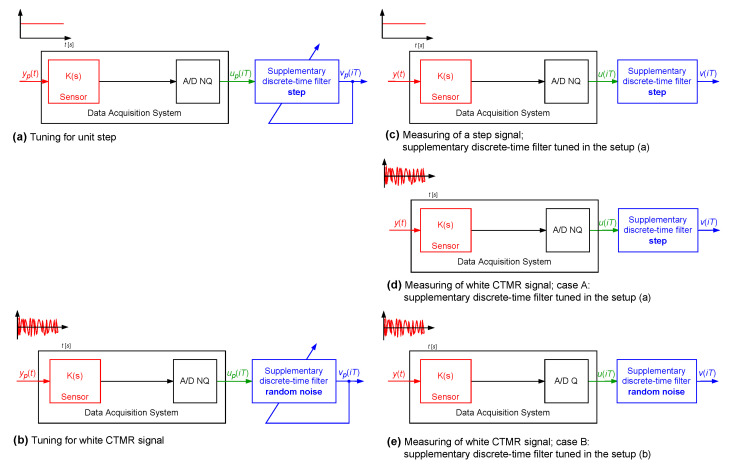
Setup of a simulation experiment—case of deterministic signal measurements.

**Figure 20 sensors-23-01676-f020:**
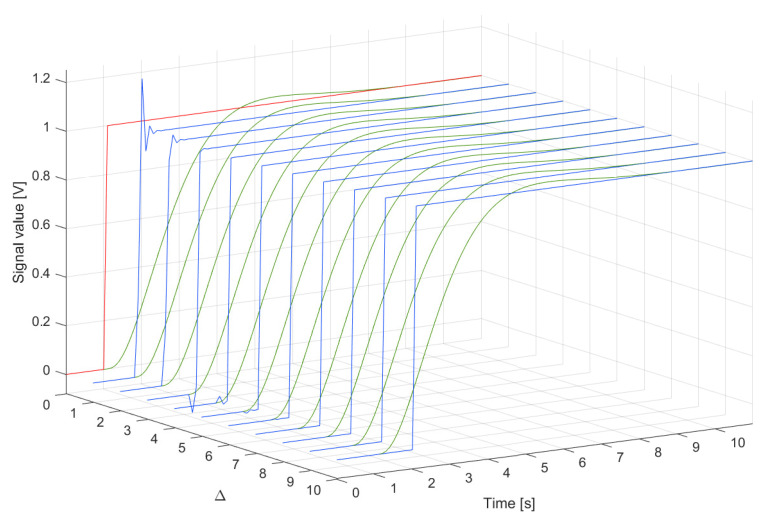
Input signal $y_{p}\left(t\right)$ (red line), response $u_{p}\left(iT\right)$ of the sensor (green lines) and responses $v_{p}\left(iT\right)$ (blue lines) of the sensor with the attached supplementary discrete-time filter for different values of Δ in the range [0,11] s, T=0.1000 s and dF=39.

**Figure 21 sensors-23-01676-f021:**
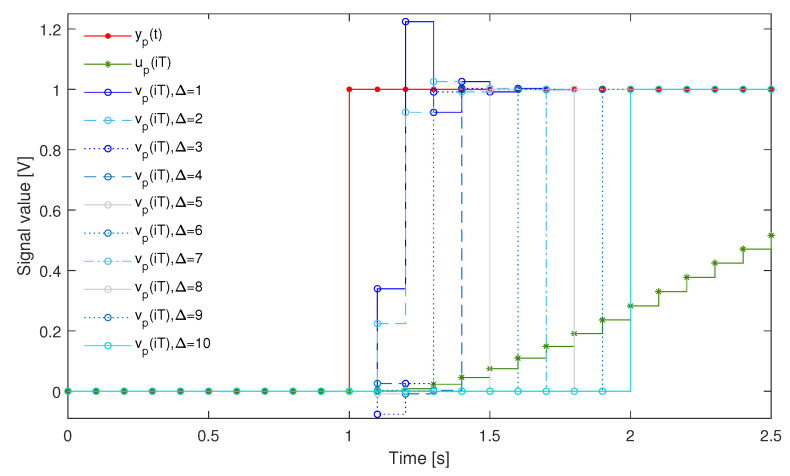
Input signal $y_{p}\left(t\right)$ (red line), response $u_{p}\left(iT\right)$ of the sensor (green lines) and responses $v_{p}\left(iT\right)$ of the sensor with the attached supplementary discrete-time filter for different values of Δ (lines in different shades of blue), T=0.1000 s and dF=39.

**Figure 22 sensors-23-01676-f022:**
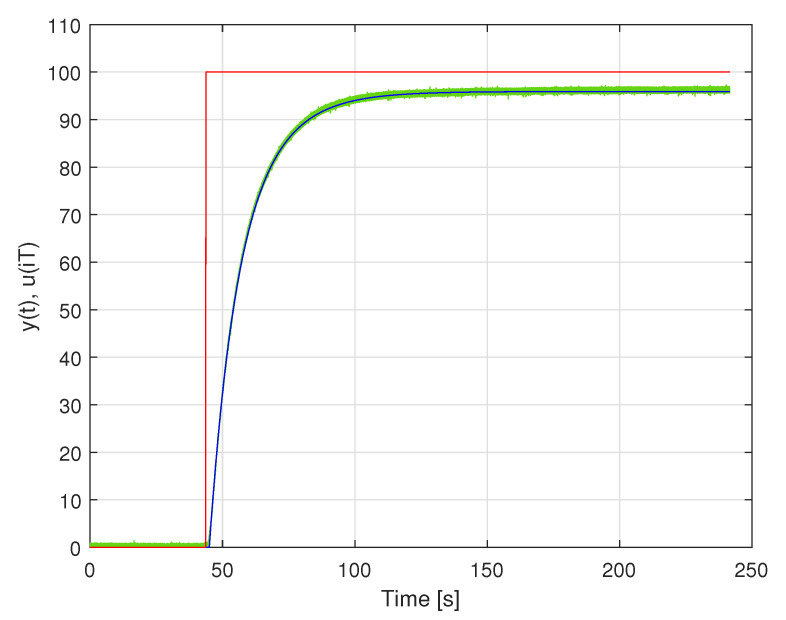
Responses of the temperature sensor (green line) and the corresponding model (blue line) to the step excitation (red line).

**Table 1 sensors-23-01676-t001:** Correction quality index values for different cases of using supplementary discrete-time filters.

Time Constant/System	L=0.0100	L=0.0500	L=0.1000	L=0.5000
no. 1	0.0012	0.1300	0.0218	0.0584
no. 2	0.0095	0.0095	0.0095	0.0095
no. 3	0.1100	0.3600	1.9700	317.7600
no. 4	0.0306	0.3000	0.1790	15.5800
no. 5	1.7700	27.8500	49.9000	83.8800

**Table 2 sensors-23-01676-t002:** Values of the cost function $S_{2}\left(\Delta \right)$ and correction quality index C(Δ,dF) for the sensor with the attached supplementary discrete-time filter (dF=39): case A—the supplementary discrete-time filter tuned for the unit step as $y_{p}\left(t\right)$; case B—the supplementary discrete-time filter tuned for a single realisation of the band limited to the range [0,5] Hz white CTMR signal as $y_{p}\left(t\right)$.

Δ	$S_{2}\left(\Delta \right)$ for the Unit Step	C(Δ,39) for a Single Realisation of the White CTMR Signal—Case A	C(Δ,39) for a Single Realisation of the White CTMR Signal—Case B
1	49	14	1.5
2	5.7	61	$1.8\cdot 10^{-1}$
3	$6.6\cdot 10^{-1}$	66	$1.5\cdot 10^{-1}$
4	$8.1\cdot 10^{-2}$	61	$1.7\cdot 10^{-1}$
5	$1.0\cdot 10^{-2}$	61	$1.7\cdot 10^{-1}$
6	$1.0\cdot 10^{-3}$	61	$1.3\cdot 10^{-1}$
7	$1.2\cdot 10^{-4}$	61	$1.5\cdot 10^{-1}$
8	$1.3\cdot 10^{-5}$	61	$1.6\cdot 10^{-1}$
9	$1.5\cdot 10^{-6}$	61	$1.6\cdot 10^{-1}$
10	$1.8\cdot 10^{-7}$	61	$1.3\cdot 10^{-1}$

## Data Availability

Not applicable.

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
