# Peer review of "Correction of Dynamical Properties of Data Acquisition Systems"

_sensors, 2023, doi:10.3390/s23031676_

Round 1

Reviewer 1 Report

This paper present smart Data Acquisition Systems. The paper was organized well but the language should be improved.

The literature review section includes old references.

Comparing the performance of the presented system from the point of view of accuracy and speed of calculation with previous works has not been done to confirm the strengths mentioned in the work.

In data acquisition systems, simplicity, speed and accuracy are three important parameters that should be evaluated in comparison with similar methods.

Author Response

Thanks very much for your comments. Please find the attachment of our responses.

Reviewer 2 Report

The manuscript proposed a new approach to the correction of dynamical properties of the data acquisition system based on an estimation of delayed signal values at its input by using a specially designed supplementary discrete-time filter. 

Reviewer 3 Report

The paper proposes an approach that can achieve accurate and fast measurements. Supplementary discrete-time filter is proposed to estimate the values of delayed samples of the measured signal.  Overall, this paper is well-written and organized. However, there are few papers referenced in the last three years in the references, and it is suggested to add the research advances in recent years.

Reviewer 4 Report

Comment

    This paper presented a new approach to correction of dynamical properties of data acquisition system based on an estimation of delayed signal values at its input by using specially designed supplementary discrete-time filter,which was illustrated with simulated case study examples, and showed effectiveness of the proposed approach. So this article has the referential Value in a certain sense. However, the following concerns and doubts need to be addressed before the paper can be further considered.

(1) How to prove the effectiveness of the proposed method for the nonlinear data acquisition systems and sensors? It is better to explain with actual cases.

(2) Please use actual cases to prove the effectiveness of the proposed method, not just simulation.

This is expectant that the manuscript would be further improved.

Reviewer 5 Report

The Authors proposed " Correction of Dynamical Properties of Data Acquisition Systems”

The authors should consider the following suggestions provided by the reviewer in order to improve the scientific depth of their manuscript, as well as they should address the following comments in order to improve the quality of the presentation of their manuscript:

1-    Check typos and grammatical errors throughout the text.

2-      In the abstract section, the result of this work must be briefly described with data (please add some numerical results (achievements) in this part).

3-      In the Introduction part, strong points of this proposed article should be further stated. Not clear what is the originality of this work? What makes it different from existing work?

4-      The novelty and major contribution of the research need to be properly described.

5-      Past studies should be reviewed well. The disadvantages of reviewed studies should be highlighted. The new features of the proposed method and the main advantages of the results over others should be clearly described. The author(s) must highlight in 8-10 lines what gaps are observed in existing literature, which has led to the design of the proposed method in this paper.

6-      The author(s) have to pay more attention to the writing of mathematical equations and their parameters.

7-      The quality of some figures is poor, the author(s) must redraw them with high quality. Some text on figures is difficult to read.

8-      More details are needed about the experiments.

9-    The results of the proposed method are not compared against other methods. Why not any recent state-of-the-art? So, how will we know about the efficacy of the proposed work with respect to recent works? This presents an important hindrance to the proposed work.

10-   Many references are out of date; they must be updated with recent ones.

Round 2

Reviewer 1 Report

The paper can be accepted in this form